# Synthesis and Characterization of Hydroxyethyl Cellulose Grafted with Copolymer of Polyaniline and Polypyrrole Biocomposite for Adsorption of Dyes

**DOI:** 10.3390/molecules27238238

**Published:** 2022-11-25

**Authors:** Majed A. Bajaber, Muhammad Naveed Anjum, Muhammad Ibrahim, Tahir Farooq, Mirza Nadeem Ahmad, Zain ul Abideen

**Affiliations:** 1Chemistry Department, Faculty of Science, King Khalid University, P.O. Box 9004, Abha 61413, Saudi Arabia; 2Department of Applied Chemistry, Government College University Faisalabad, Faisalabad 38000, Pakistan

**Keywords:** biocomposite, adsorption, dyes removal, pollution

## Abstract

The emerging concepts of sustainable textiles and controlled production strategy demands ideally zero emissions of contaminants into the aquatic environment. However, the currently in-practiced conventional processes in textiles dispose of a number of contaminants especially super toxic synthetic dyes as effluents. In recent years, nanomaterials have become attractive choice for eco-friendly removal of organic dyestuff. Accordingly, this article reports synthesis and characterization of biocomposite wherein copolymer of polyaniline (PANI) and Polypyrrole (PPY) was grafted onto hydroxyethyl cellulose (HEC). Further, adsorption properties of as-prepared composite were evaluated using textile dyes Rhodamine B (RhB) and methyl Orange (MO)- as model adsorbate. The characterization of novel biocomposite (HEC/PANI-PPy) was carried out using Fourier Transform Infrared (FT-IR), Brunauer-Emmett-Teller analyzer (BET), Scanning Electron Microscope (SEM), and powder X-ray diffraction (XRD). The operational parameters such as dye initial concentration, adsorbent amount, pH and contact time were also studied to evaluate the efficiency level of the prepared biocomposite. Interestingly, the composite-mediated adsorption of RhB and MO followed pseudo-second order and the Langmuir isotherm. It is found that the adsorption capacity HEC/PANI-PPy is 30.06 and 29.3 for RhB and MO respectively. Thus, HEC/PANI-PPy is an inexpensive and highly efficient adsorbent that could be employed for could be employed for the separation and removal of toxic organic dyes from polluted textile effluents.

## 1. Introduction

Generally, the wastewater effluents released from textile industries contain heavy loads of pollutants majorly including synthetic dyes. Organic dyes have complex structures with non-biodegradable nature damaging the taste and quality of groundwater. They disturb the oxygen transfer mechanisms, light penetration, and particulate contents in water bodies. Thus they become toxic to the environment, hazardous to aquatic life and harmful to growing plants [1]. The dyestuffs owing to their persistent nature and high aqueous stability could become part of the food chain ultimately posing severe threats to human beings [2,3]. Therefore, over the last decade, the awareness of the concepts of sustainable textiles and controlled production strategy demands ideally zero emissions of contaminants from industry into the aquatic environment. However, the currently in-practiced conventional processes in textile dyeing are inefficient and unable to control the discharge of dye-enriched effluents. Since the zero-emission cannot be validated with the existing infra-structures and traditional dyeing technologies, a number of chemical, biological and physical approaches have been employed for effluent treatments for the removal of dyes and other pollutants [4,5,6]. Among physical treatment methods, the adsorption methodology has received much industrial attention due to cheaper infrastructure, high reusability, and operationally facile approaches [7,8,9]. Over the years various adsorbents with different physical and chemical properties have been utilized for dye removal by adsorption showing different efficiency levels. Therefore, there has always been desirous to introduce new adsorbents with high efficiency and reusability at low cost.

In recent years, the introduction of nanomaterials as highly efficient adsorbents for organic pollutants has revolutionized the scope of contaminant remediation and environment management [10,11]. In this connection, biocomposites synergizing high adsorption properties with photocatalytic activities proved their worth as highly efficient adsorbents for the removal and degradation of organic dyes [12,13]. The surface functionalization of biopolymers capacitates them to interact easily with dyes and heavy metals thus, highlighting them as ideal candidates for the development of new adsorbents [14,15]. The cellulose derivatives especially hydroxmethyl cellulose (HEC) have shown promising adsorption properties for the removal of heavy metals and other contaminants from wastewater effluents [16,17]. HEC has biodegradable and biocompatible nature along with superior chemical stability and water solubility thus, an excellent choice for the development of highly efficient new adsorbent.

In recent years, conducting polymers in general and polyaniline and polypyrrole in particular, have attracted much attention due to their versatile applications, such as electrochemical devices, sensors, electromagnetic shielding batteries, water treatment, supercapacitors, etc. [18,19,20,21,22]. In addition to this, polyaniline and polypyrrole have also been employed as adsorbents in elimination of toxic metals and dyes from aqueous media [23,24,25,26,27]. The selection of polyaniline and polypyrrole in present study was triggered by their low cost, ease of synthesis in bulk, non-toxicity, biocompatibility, and large specific surface area [25,28,29,30,31,32]. Furthermore, the presence of a lone pair of electrons on nitrogen atom in PPY and PANI facilitates adsorptive interaction (π-π interaction and hydrogen bonding) with dye in polymeric chain (Figure 1). In this connection, limited or no work has been reported on the removal of dyes using the PANI–PPY copolymeric composite with biopolymer as an adsorbent. In short, our team reports the synthesis of hydroxyethyl cellulose (HEC) grafted with a copolymer of polyaniline (PANI) and polypyrrole (PPy) and its successful use for the adsorption of RhB and Methyl orange. The adsorption capacities of some famous adsorbents for MO and RhB see Table 1.

## 2. Materials and Methods

### 2.1. Chemicals

Hydroxyethyl cellulose (HEC) was supplied by a Sigma-Aldrich and was used as received. Methyl Orange (MO), Rhodamine B (RhB) and ammonium persulphate (APS) were purchased from Hongye Chemical and Aladdin Chemical Reagent corporation Shanghai, China respectively. Diaminodiphenylaniline (DDPA) bought from TCI, Shanghai, China. Figure 1 and Figure 2 represent schematic illustration of chemical structures and grafting onto HEC via polymerization. The structures of dyes have been shown in Figure 3 [19] and Figure 4 [20].

The UV-Visible spectrophotometer (Lambda 25, Perkin Elmer, Shanghai, China) was employed for measuring maximum wavelength λ_max_ and absorbance of dyes samples before and after adsorption.

### 2.2. Preparation of HEC/PANI-PPy

Biocomposite was prepared by mixing Hydroxyethyl cellulose (HEC) solution and Monomers solution as represented in scheme (Figure 2). Hydroxyethyl ce2.3llulose (HEC) was prepared by dissolving 0.2 g of hydroxyl ethyl cellulose (HEC) in 0.1 N HCl solution (Prepared by dissolving 8.3 mL of 37% HCl (12 M) in 1 L of distilled water) and stirred it for 15 min. The monomers’ and DDPA dimer solution was prepared by dissolving 0.18 g Aniline, 0.36 g Pyrrole and 0.005 g DDPA in 0.1 N HCl solution and stirring it for 15 min. After preparing both solutions poured both solutions into a common beaker and stirred them continuously for 3 h at room temperature meanwhile adding 0.1 M Ammonium persulfate (APS) solution mix with 20 mL 0.1 N HCl solution dropwise. HEC/PANI-PPy so prepared, was centrifuged, washed using distilled water, dried in oven around 60 °C for 3 h, and stored[21].

### 2.3. Characterization of HEC/PANI-PPy

The Fourier transform infrared (FT-IR) spectra were taken using Spectrum 2, Perkin Elmer, Shanghai, China. between the range of 400–4000 cm^−1^. The morphology of HEC/PANI-PPy was examined using scanning electron microscope (SEM). The X-ray diffraction (XRD) patterns of HEC/PANI-PPy were recorded using X-ray spectrometer (D8 Advance Bruker, Billerica, MA, USA). The surface area analyses were carried out using Brunauer-Emmett-Teller (BET) by adsorption of gas molecules on the surface of HEC/PANI-PPy. 

## 3. Results and Discussion

### 3.1. Fourier Transform Infrared Spectroscopy (FTIR)

The FT-IR spectra of pure HEC (Figure 5) showed clear differences from that of HEC/PANI-PPy (Figure 6). Figure 5 of HEC spectrum showed a broad absorptions at 3358 cm^−1^ and can be due to “O-H stretching” vibrations [22]. Another “medium absorption band” is present at 2875.57 cm^−1^ allocated for the C-H stretching vibrations. More peaks i.e., 1407 cm^−1^ and 1369 cm^−1^ can be assigned to O-H “plane deformation” and for C-H “symmetric bending vibrations” respectively in -CH_2_O- [23]. Moreover, beta-(1,4) glycoside linkage was indicated by absorption band of present at 889 cm^−1^. The absorption peak at 1052 cm^−1^ is due to C-O anti-symmetric vibrations [24]. Similarly, the absorption peak at 1020 cm^−1^ owes to the C-O-C stretching vibrations in the glucopyranose structure [25]. On the other hand, FT-IR spectrum of HEC/PANI-PPy (Figure 6), a true difference can be observed in the spectrum of pure HEC and HEC/PANI-PPy. In comparison, the spectrum of HEC/PANI-PPy, the absorption of peaks at 3399 cm^−1^ and 3199 cm^−1^ are due to the O-H and N-H “stretching vibrations” respectively [26]. The absorption peaks of 3053 cm^−1^ and 2874 cm^−1^ are due to C-H stretching vibrations. Grafting of PANI and PPy onto HEC is represented by the characteristic peaks which are obtained at1637 cm^−1^ for C=C and 1432 cm^−1^ for C=N [27]. Moreover, the aromatic stretching and C-C bending vibrations appeared at 1551 cm^−1^ and 1497 cm^−1^ respectively [28]. Band at 1175 cm^−1^ is due to the carbonyl group of cellulose [29]. The series of peaks present at 733 cm^−1^, 887 cm^−1,^ and 1043 cm^−1^ are different vibrational modes of PPy [30].

### 3.2. X-rays Diffraction (XRD) Analysis

XRD was used to analyze the nature of biocomposite. HEC and HEC/PANI-PPy were analyzed to check the amorphous or crystalline nature of the HEC/PANI-PPy. [31]. XRD is extensively used to elaborate the nature of the polymeric materials. According to Figure 7, the sharp and narrow peaks at 2θ = 24.10°, 26.70°, 30.90°, and 33.10° suggested the crystalline HEC/PANI-PPy. These findings are in consistency with the previously reported the crystalline nature of the biocomposite [26]. Figure 7 showed the XRD pattern of HEC/PANI-PPy. This result showed that PANI and Ppy are successfully grafted on the HEC as the peaks of PANI and Ppy are confirmed from the previous studies. PANI showed a peak at 2θ = 22.08° and Ppy at 2θ = 26.81° [32]. The pattern of Figure 7 XRD showed that HEC/PANI-PPy has crystalline nature while Figure 8 XRD pattern is indicating that HEC has amorphous nature as it has no sharp peaks [40].

### 3.3. Morphological Study by Scanning Electron Microscopy (SEM)

The Figure 9 and Figure 10 shows the morphological pictures of the HEC and HEC/PANI-PPy and it is noted in the figure that the hydroxyethyl cellulose fibers are highly agglomerated and thread-like appearances. The SEM images of HEC shown in the Figure 9 and its corresponding Biocomposite present in Figure 10 shows that the presence of PANI and PPy covers the HEC morphology. Due to the high amount of synthetic polymer content in the Biocomposite, both HEC surface and pores grafted with PANI and PPy and created a rough and granular structure that imparts a crystal nature for adsorption [41].

### 3.4. Brunauer-Emmett-Teller (BET)

BET isotherm graph shown in the Figure 11 indicated that the material is porous. The deflecting curve at the start is indicating that the adsorption phenomenon is monolayer and the steady curve is showing that pores are filling while increasing the pressure but at the end, the adsorption of gas increased on the adsorbent indicating the multilayer adsorption as well as desorption of the gas molecules [42].

### 3.5. Adsorption of Dyes

In these adsorption experiments, we used MO and RhB since these dyes are frequently used in textile industries and for many other applications. We conducted all experiments at room temperature. The aqueous solution of MO and RhB are having a concentration of 500 mgL^−1^ by dissolving 0.5 g of the dye under examination into 1 L of double distilled water. Dyes’ solutions in water were obtained from stock solution by diluting to get desired concentrations (1.25–30 mgL^−1^). Calibration curves of both dyes were drawn by plot of absorbance versus the concentration of MO and RhB which are based on the UV-visible spectra of adsorption of the standard solutions (1.25–30 mgL^−1^). To study the kinetics of adsorption, 10 mg of HEC/PANI-PPy was added to 40 mL (30 mgL^−1^) of MO and RhB solutions respectively. A fixed amount (2 mL) of both the mixtures was taken out at predefined time span followed by centrifugation at 5000 rpm for five minutes one by one of both solutions. The remaining concentration of MO, RhB, and remaining solution was checked with a UV-Visible spectrophotometer. The contact time during isotherm experiments, was kept 2 h. The amounts dyes adsorbed onto biocomposite were calculated according to the following equation.
$$Q_{e}=\left(C_{o}-C_{e}\right)\;V/m$$
where *m* is the mass of added adsorbent, *Q_e_* is the amount of dye absorbed on one gram of absorbent (mgg^−1^) at point of equilibrium, *C_o_* and *C_e_* are the initially taken and equilibrium concentrations of dyes (mgL^−1^) respectively and *V* is the volume of solution [43].

## 4. Dyes Adsorption and Effect of Different Parameters

### 4.1. Effect of pH

In general, pH of dye solution depends on degree of dissociation of adsorbent and its active sites [44]. Previous studies revealed that charge on the surface of adsorbent plays a vital role during adsorption [45]. The adsorption capacities of dyes under examination were checked by varing the pH from 2.0–11 at the initial concentration of dye i.e., 50 ppm and 2 h interaction time. Figure 12 shows the effect pH on adsorption of Methyl orange and Rhodamine B. Further adsorption studies of both dyes are carried out at optimum pH.

### 4.2. Effect of Temperature

Industrial wastewater is mostly disposed of at higher temperatures. In order to check the effect of dyes’ solution temperature on the adsorbing capacities of dyes by HEC grafted with a copolymer of Polypyrrole and polyaniline, the temperature of solution was varied from 302–337 K while other parameters kept at optimum.

Like many physical properties such as adsorption of dyes is also affected by rise and fall of temperature and adsorption can be controlled by temperature [46]. The results revealed that the adsorption capability of adsorbents was decreased with increasing temperature, showing that adsorption operation of dyes onto HEC/PANI-PPy is an exothermic process (Figure 13). It was observed that dyes showed the highest adsorption capacity at 302 K. Rise in Temperature may cause results in enlargement of pores that may cause the evasion of dyes’ molecules from adsorbent surface [47].

### 4.3. Effect of Adsorbent Dose

In order to find the effect of adsorbent dose, the amount of HEC/PANI-PPy was varied between 0.05/50 mL and 0.3 g/50 mL and 50 ppm of dye solution (Figure 14). The optimal condition remained same i.e., pH, contact time (2 h), temperature (27 °C) and 150 rpm shaking speed. It was found that adsorption ability decreased with rise of HEC/PANI-PPy which is ascribed to aggregation of the adsorbent with concomitant rise in adsorbent concentration and decreasing availability of surface for adsorption [48]. 

### 4.4. Effect of Contact Time

The solutions of MO and RhB with adsorbent HEC/PANI-PPy were separately agitated for two hours to determine the relationship between contact duration and adsorption of dye onto HEC/PANI-PPy. These contact time experiments were carried out at optimized conditions as mentioned above.

Figure 15 shows that the rate of adsorption at the beginning of the experiment was faster and decreased with the passage of time and equilibrium achieved between 20 to 50 min. However, on further increasing of time there was no considerable change in dye adsorption which may be due to rapid external surface adsorption and fast diffusion at the boundaries and slows down with the passage of time [49].

### 4.5. Effect of Initial Dye Concentration

Relationship between Adsorption capacity and initial concentration of dye was investigated by changing initial dye concentration 5–400 ppm keeping all other factors at optimum level as mentioned above. It was revealed that increasing initial concentration of dye enhanced the adsorption capacity (Figure 16). This remarkable enhancement in adsorption can be ascribed to high initial concentration of dye ions to bind with the active sites of adsorbent which ultimately resulted in increased adsorption [50].

### 4.6. Adsorption Kinetics and Isotherms

Different parameters of adsorption were studied using advanced techniques. UV-Visible plots of RhB and MO standard aqueous solutions with different concentrations from 1.25 mg/L to 30.0 mg/L were drawn. Based on absorbance values, straight lines were obtained as shown in the Figure 17 and Figure 18.

Kinetic curve for removal of RhB and MO by HEC/PANI-PPy, showed that amount of dye adsorbed (q) in the initial minutes was fast but slowed down until it attains equilibrium state (Figure 19).

Since, adsorption kinetic is referred to the measurement of adsorption uptake and diffusion of adsorbate in the pores for a particular time span at constant pressure. Further elaboration of adsorption of dyes on the surface of HEC/PANI-PPy by using the “pseudo-first order Equation (1)” and “pseudo-second order Equation (2)” [37].

We will elaborate on the

“Pseudo first order equation:(1)$$log\;log\;\left(q_{e}-q_{t}\right)=logq_{e}-\frac{k_{1}t}{2.303}$$

Pseudo second order equation”:(2)$$\frac{t}{q_{t}}=\frac{1}{k_{2}q_{e}^{2}}+\frac{t}{q_{e}}$$
where $q_{e}$ and $q_{t}$ are the amounts of dye adsorbed (mg g^−1^) onto the surface at equilibrium in time *t* (min), respectively, and $k_{1}$ is rate constant (g mg^−1^ min).

Values of R^2^ of MO and RhB for the pseudo-first order kinetic model is 0.96842 and 0.84122 respectively. The Values of R^2^ of MO and RhB for the pseudo-second-order kinetic model are 0.99966 and 0.99919 respectively. Results in the above graphs are showing that the adsorption process of MO and RhB on HEC/PANI-Ppy followed the pseudo-second-order kinetics (Figure 20 and Figure 21). Figure 22: Langmuir adsorption isotherm of MO and RhB. Figure 23. Freundlich adsorption isotherm of MO and RhB. Figure 24. Temkin adsorption isotherm of MO and RhB. The experimental equilibrium adsorption data were be further analyzed by applying isothermal models (Figure 22, Figure 23 and Figure 24) i.e., the Langmuir (3), Freundlich (4), Temkin (5) and Equation (6):
(3)$$\frac{C_{e}}{q_{e}}=\frac{1}{q_{m}K_{L}}+\frac{C_{e}}{q_{m}}$$
(4)$$logq_{e\;}=log\;logK_{F}+\frac{1}{n}\;logC_{e}$$
(5)$$q_{e}=\frac{RT}{b}ln\left(A_{T}C_{e}\right)$$
(6)$$q_{e}=a+b\;log\;log\;C_{e}$$

Plots of these isotherms reveals that the adsorption mechanism of RhB and MO onto HEC/PANI-PPy can be best fitted to Langmuir adsorption isotherm model.

### 4.7. Adsorption Mechanism

Adsorption of dyes (RhB and MO) onto the HEC/PANI-PPy is due to the π-π interaction between localized aromatic π-electron of hyper-crosslinked polyaniline and Polypyrrole and aromatic ring of dye molecules. Moreover, adsorption of dyes is also aided by the overall π-π interaction and hydrogen bonding between nitrogen atom of dye and hydrogen atom of amine groups of polymeric chain of polyaniline and (Figure 1).

## 5. Conclusions

Herein, synthesis and characterization of novel biocomposite HEC/PANI-PPy have been reported followed by its application for adsorption of toxic organic dyes. The different parameters such as initial concentration of dye, adsorbent amount, pH temperature, and contact time were investigated to establish the efficiency level of prepared composite. The adsorption mechanism of RhB and MO followed pseudo-second order and the Langmuir isotherm model. It is concluded that the HEC/PANI-PPy is a highly efficient adsorbents for removal of organic dyes and its application may possibly be extended for the removal of other pollutants from textile effluents.

## Figures and Tables

**Figure 1 molecules-27-08238-f001:**
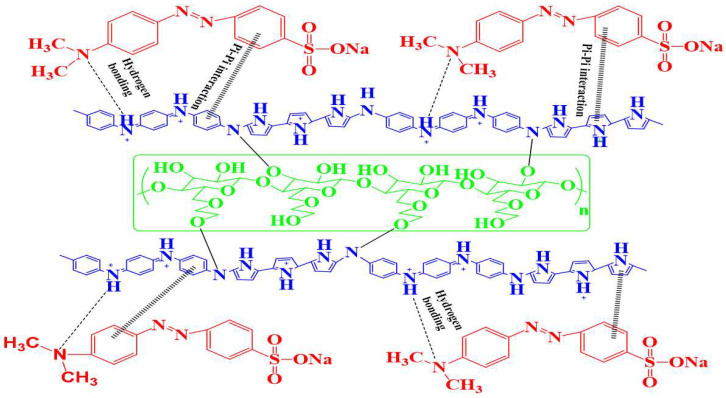
Schematic representation of HEC/PANI-PPy and dye adsorption.

**Figure 2 molecules-27-08238-f002:**
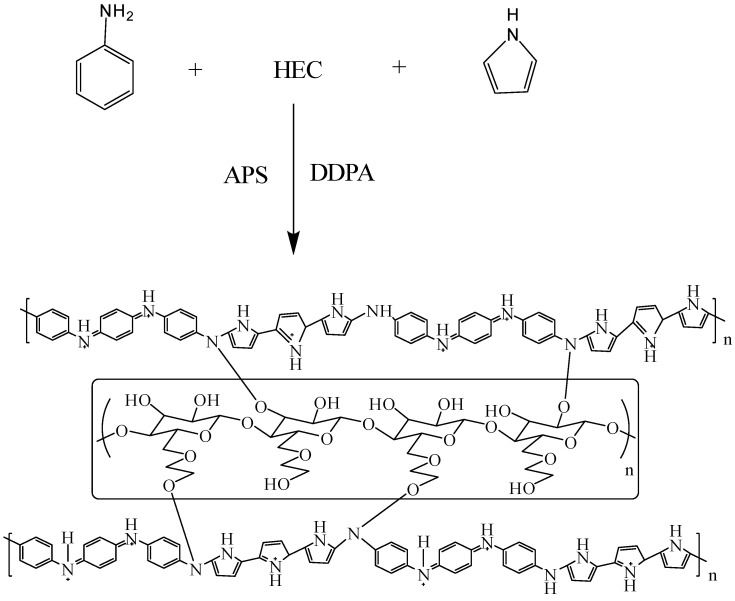
View of chemical polymerization of PANI and PPy on HEC.

**Figure 3 molecules-27-08238-f003:**
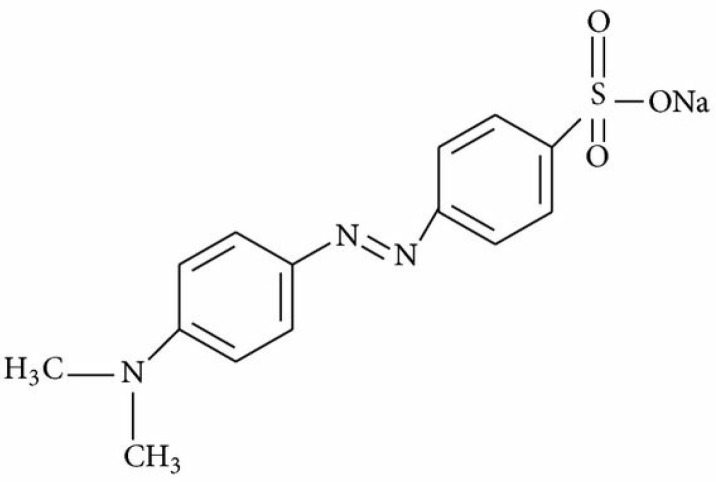
Methyl orange.

**Figure 4 molecules-27-08238-f004:**
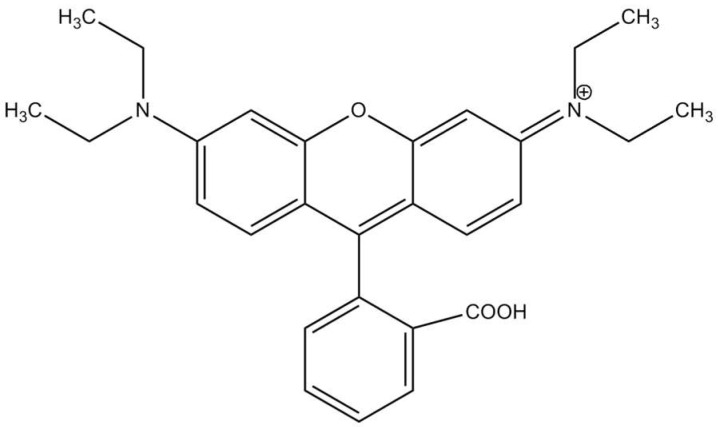
Rhodamine B.

**Figure 5 molecules-27-08238-f005:**
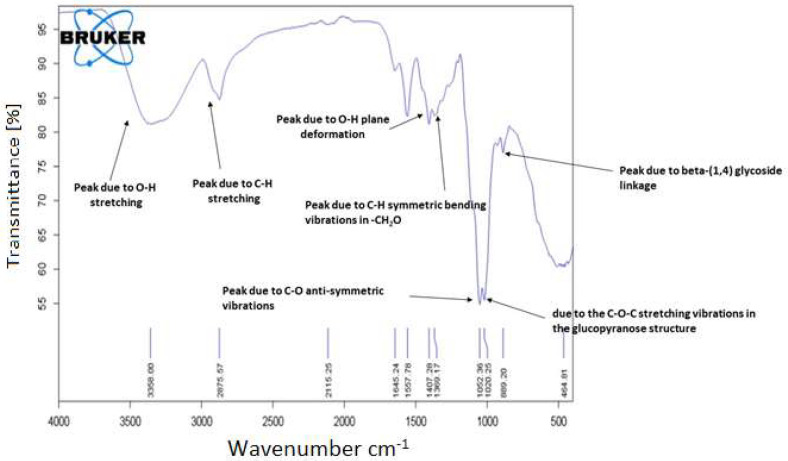
FT-IR spectrum of HEC.

**Figure 6 molecules-27-08238-f006:**
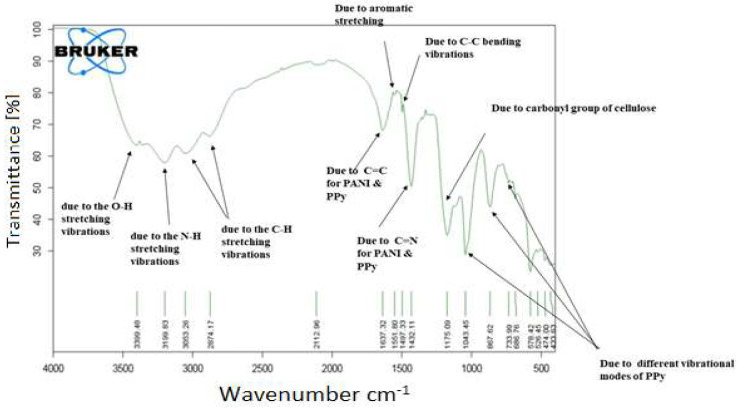
FT-IR spectrum of HEC/PANI-PPy.

**Figure 7 molecules-27-08238-f007:**
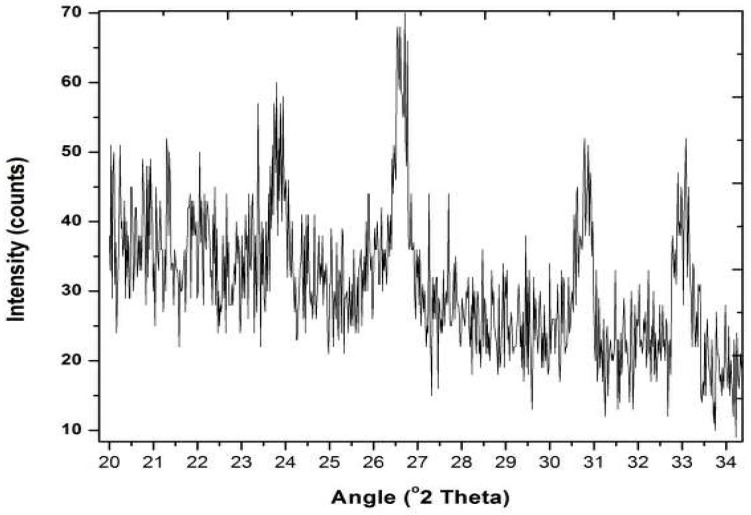
XRD of HEC/PANI-PPy.

**Figure 8 molecules-27-08238-f008:**
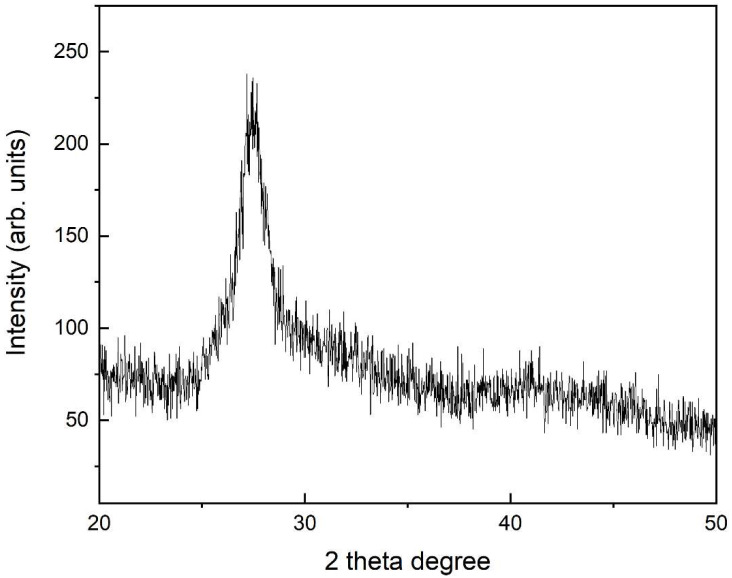
XRD of HEC.

**Figure 9 molecules-27-08238-f009:**
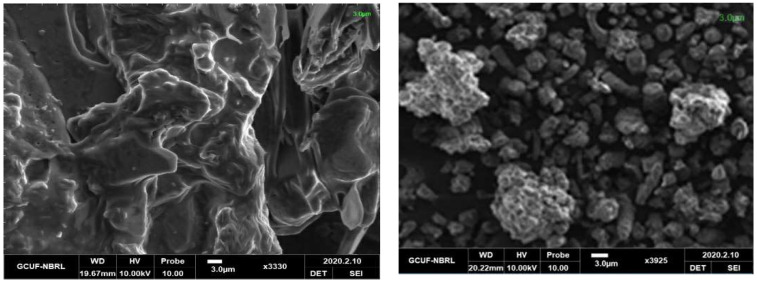
SEM of HEC.

**Figure 10 molecules-27-08238-f010:**
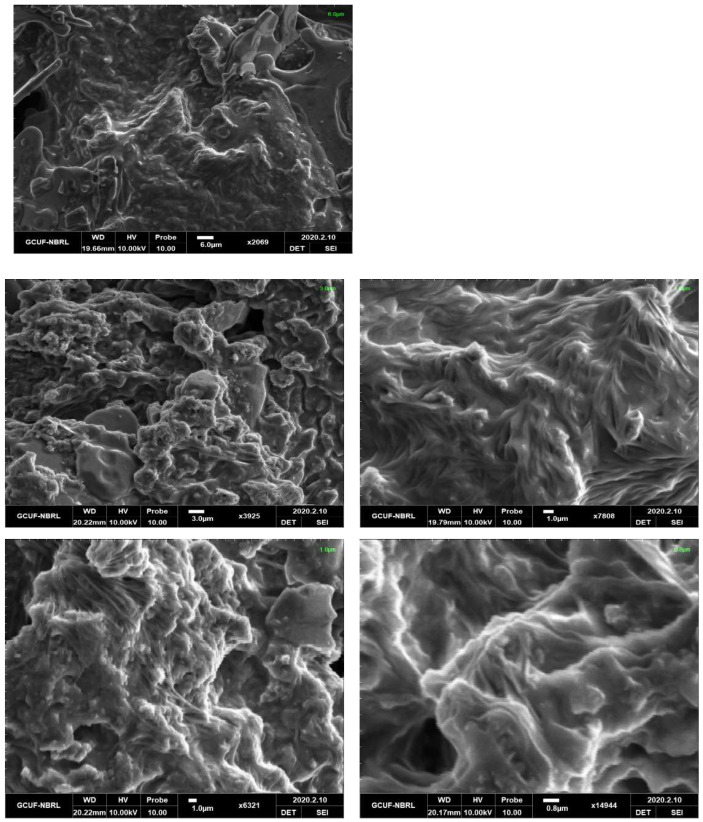
SEM images of HEC/PANI-PPy.

**Figure 11 molecules-27-08238-f011:**
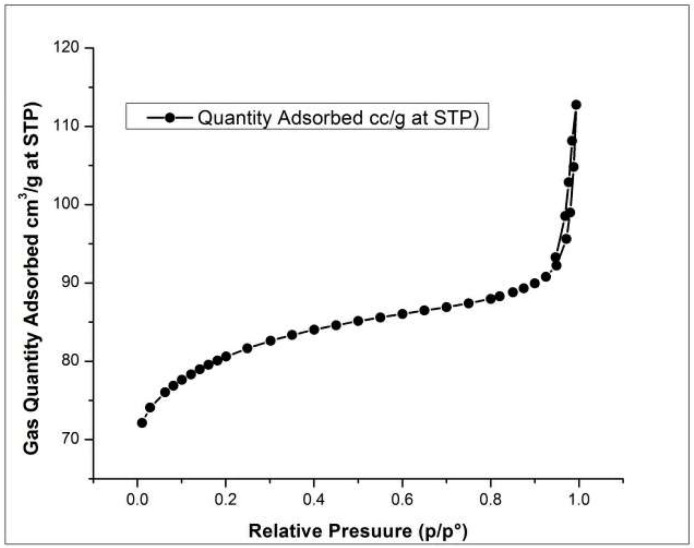
BET of HEC/PANI-PPy.

**Figure 12 molecules-27-08238-f012:**
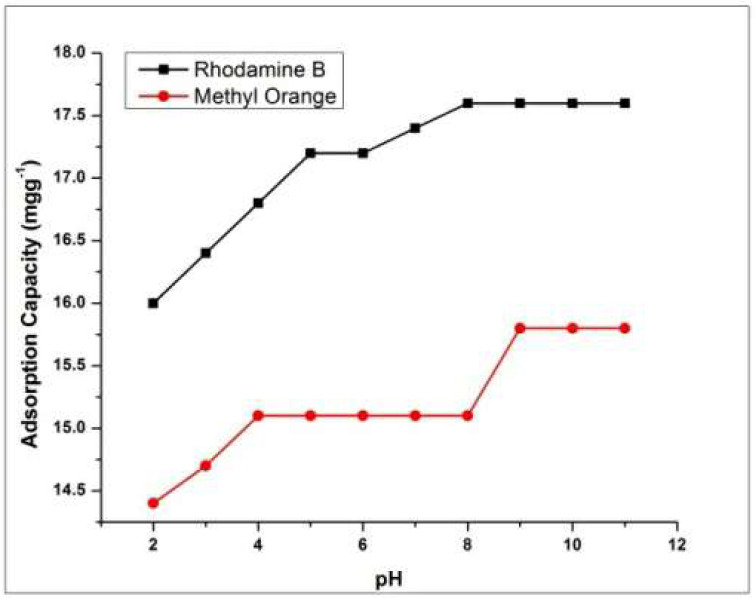
Effect of pH on dyes adsorption.

**Figure 13 molecules-27-08238-f013:**
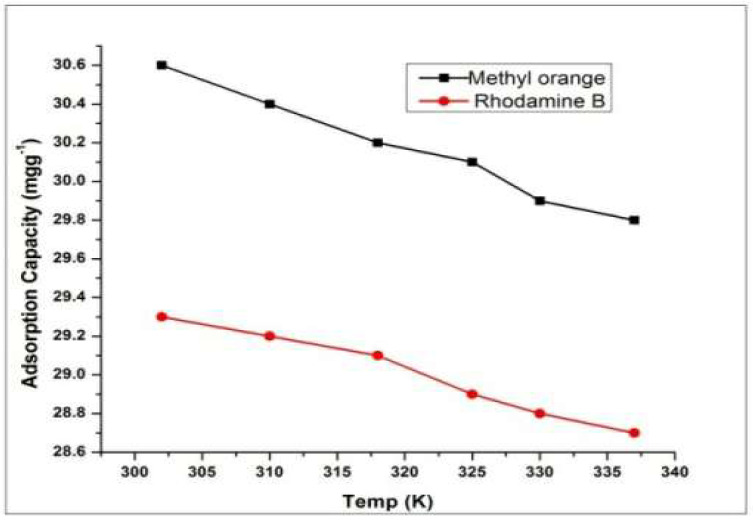
Effect of temperature on dyes adsorption.

**Figure 14 molecules-27-08238-f014:**
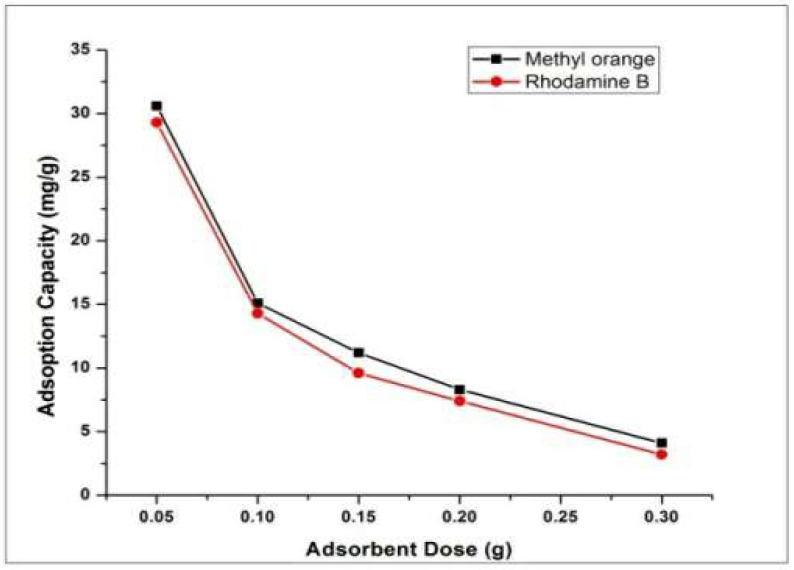
Effect of adsorbent dose on dyes adsorption.

**Figure 15 molecules-27-08238-f015:**
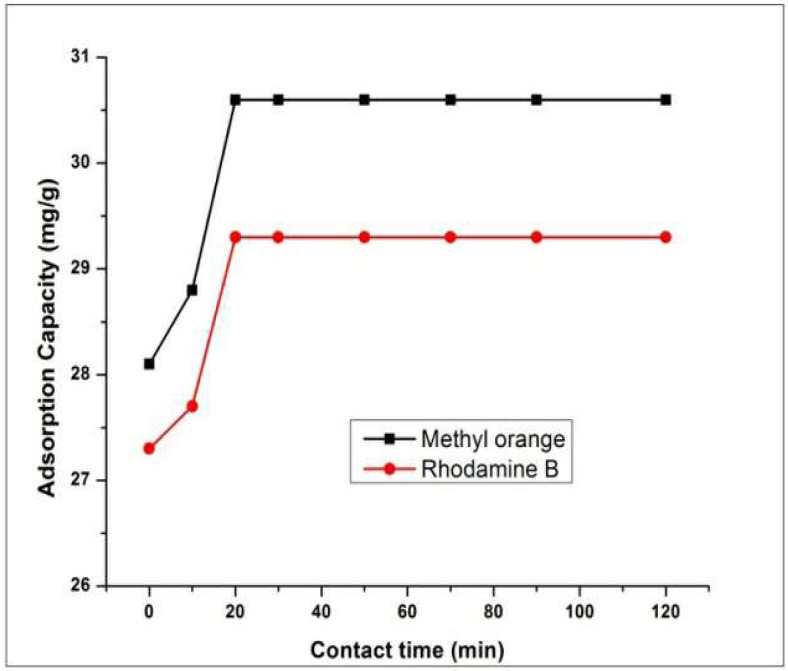
Effect of contact time on dyes adsorption.

**Figure 16 molecules-27-08238-f016:**
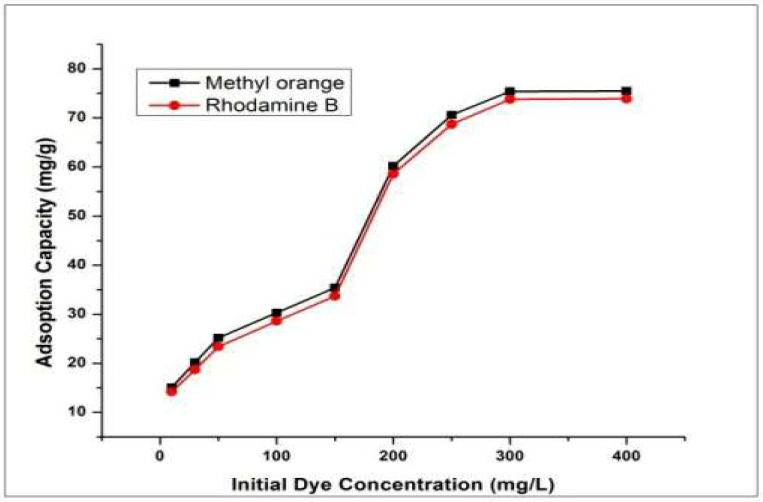
Effect of initial dye concentration on dyes adsorption.

**Figure 17 molecules-27-08238-f017:**
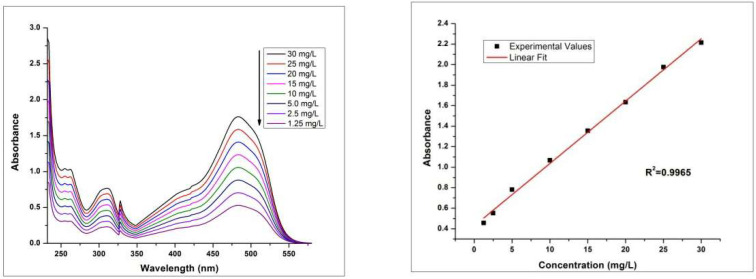
UV-Vis plot of standard aqueous solutions of MO for different concentrations (**left**) and the linear fit standard curve (**right**).

**Figure 18 molecules-27-08238-f018:**
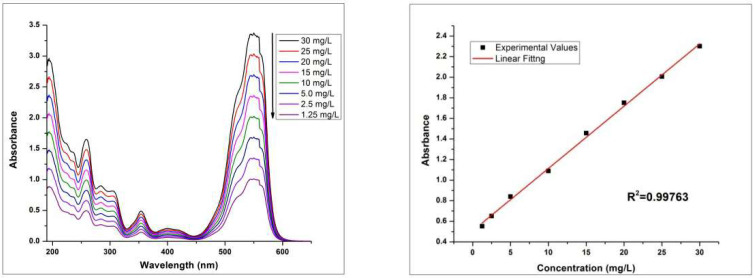
UV-Vis plot of standard aqueous solutions of RhB for different concentrations (**left**) and the linear fit standard curve (**right**).

**Figure 19 molecules-27-08238-f019:**
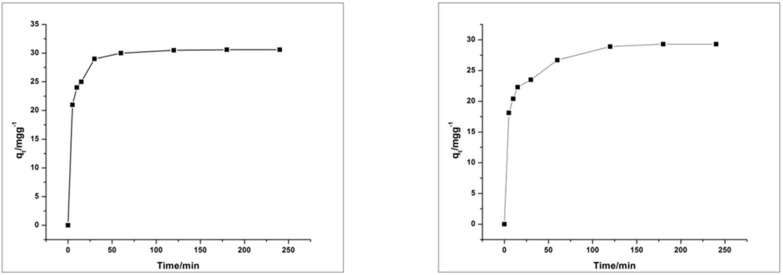
Adsorption kinetic curve for MO (**left**) and RhB (**right**).

**Figure 20 molecules-27-08238-f020:**
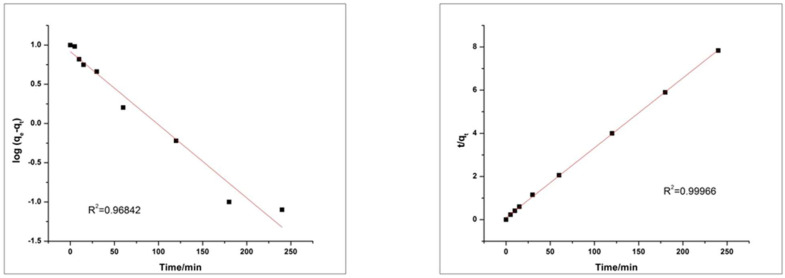
Pseudo first order (**left**) and Pseudo second order (**right**) kinetic plot for MO.

**Figure 21 molecules-27-08238-f021:**
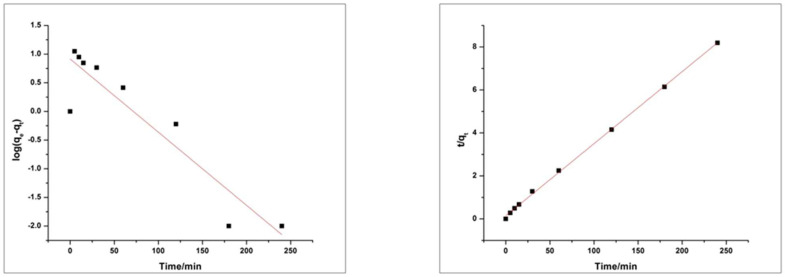
Pseudo first order (**left**) and Pseudo second order (**right**) kinetic plot for RhB.

**Figure 22 molecules-27-08238-f022:**
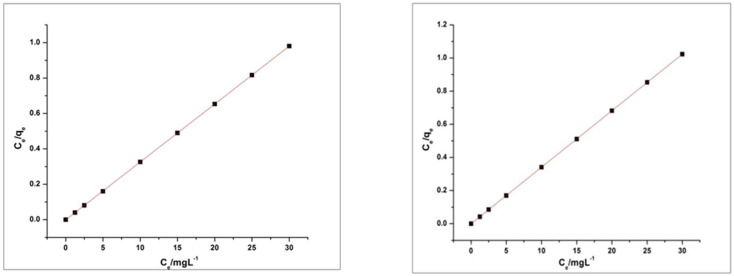
Langmuir adsorption isotherm of MO and RhB.

**Figure 23 molecules-27-08238-f023:**
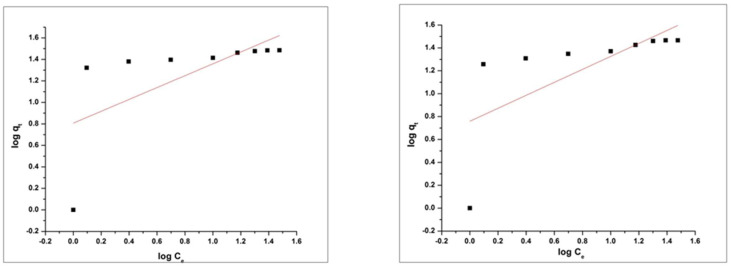
Freundlich adsorption isotherm of MO and RhB.

**Figure 24 molecules-27-08238-f024:**
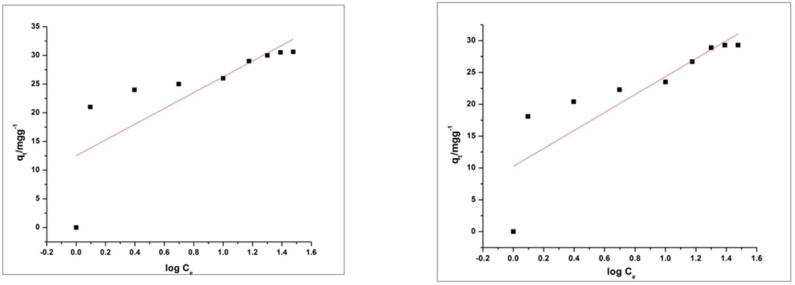
Temkin adsorption isotherm of MO and RhB.

**Table 1 molecules-27-08238-t001:** Adsorption capacities of some famous adsorbents for MO and RhB.

Adsorbent	Dye	Adsorption Capacities (mg/g)	References
MIL-125 (Ti)	RhB	59.92	[33]
CZIF-867	RhB	119.9	[34]
Ni@MOF-74(Ni)	RhB	209.2	[35]
POM@UiO-66	RhB	225.7	[36]
POC porous microspheres	MO	35.21	[37]
Banana peel	MO	21.0	[38]
COMRC	MO	34.48	[39]
Nano-composite film Maghemite/chitosan	MO	29.41	[18]
HEC/PANI-PPy	MO	30.6	This work
HEC/PANI-PPy	RhB	29.3	This work

## Data Availability

Not applicable.

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
