# Peer review of "Synthesis and Characterization of Hydroxyethyl Cellulose Grafted with Copolymer of Polyaniline and Polypyrrole Biocomposite for Adsorption of Dyes"

_molecules, 2022, doi:10.3390/molecules27238238_

Round 1

Reviewer 1 Report

In this study, the authors proposed the novel biocomposite (HEC/PANI-PPy) as adsorbent to be distinguished compared to other adsorbent on dyes adsorption. However, this study suffers from proper and deep interpretation of the goals of this study.

1. Why the authors combined HEC with two polymers of PANI and PPy? How about contribute of HEC, HEC/PANI, and HEC/PPy on the adsorption of dyes?

2. The authors characterized the HEC/PANI-PPy using BET is responsible for the adsorption in performance. Did the authors measure the surface area of the HEC/PANI-PPy?

3. The fig. 5, fig.6, fig. 7, and fig. 8 are not clearly drawn.

4. The contact time of adsorption process in line 184 was kept at 3 hrs, why the authors used 2 hrs on the effect of pH and the effect of adsorbent doses, the also the stirring rate?

5. Although several precise analytical procedures were applied there is a lack of serious discussion supported by obtained data describing the mechanism of dyes adsorption by prepared the HEC/PANI-PPy composite.

6. How the pH is controlled at certain point to investigate the effect of pH, since the pH is important parameter to know the adsorption mechanism in this research. Is that the initial pH or equilibrium pH on the adsorption processes? The Authors should explain more even in the others effect on the adsorption process.

7. What is the electrical charge of surface sites of the HEC/PANI-PPy at different pH values?

8. The authors state that increasing the temperature will decrease the adsorption capacity, indicate the endotherm process, but did not support the ∆H and others thermodynamic data. What is the possibility of chemical reaction and or ion exchange involve besides adsorption process?

9. How about the reusability of this HEC/PANI-PPy composite in adsorption-desorption process of dyes? What was the desorption efficiency (%)? This should be at least mentioned in the text. Would the same desorption method be used for dye degradation and reuse, or would a specific desorption method need to be developed for each contaminant?

Author Response

We would greatly thank all reviewers for your very helpful comments. After carefully revised our manuscript according to your comments, we have found that our conclusions become more considerate and clearer. The detailed responses to your comments are listed below.

General comment: In this study, the authors proposed the novel biocomposite (HEC/PANI-PPy) as adsorbent to be distinguished compared to other adsorbent on dyes adsorption. However, this study suffers from proper and deep interpretation of the goals of this study.

Answer: Thank you very much for your favorable recommendation of our work. We have tried to further improve the quality of our manuscript by revising it according to your convincing comments.

Comment 1: Why the authors combined HEC with two polymers of PANI and PPy? How about contribute of HEC, HEC/PANI, and HEC/PPy on the adsorption of dyes? 

Answer: Very good question. We are working on various natural biopolymers such as chitin, chitosan, cellulose and derivatives thereof and effect of conducting polymers towards adsorption of dyes and drugs such as chitosan grafted with polyaniline (Open Chemistry 2020; 18: 843–849). In future we shall report a comparative study on the effect of various polymers and copolymers.

Comment 2: The authors characterized the HEC/PANI-PPy using BET is responsible for the adsorption in performance. Did the authors measure the surface area of the HEC/PANI-PPy?  

Answer: Very relevant question. We have added data of surface area in the manuscript.

Comment 3: The fig. 5, fig. 6, fig. 7 and fig. 8 are not clearly drawn.

Answer: Figures fig. 5, fig. 6, fig. 7 and fig. 8 have been improved.

Comment 4: The contact time of adsorption process in line 184 was kept at 3 hrs, why the authors used 2 hrs on the effect of pH and the effect of adsorbent doses, the also the stirring rate?

Answer: Time 3hrs is mentioned in preparation and drying of HEC/PANI-PPy and 2hrs on the effect of pH and the effect of adsorbent doses.  

Comment 5: Although several precise analytical procedures were applied there is a lack of serious discussion supported by obtained data describing the mechanism of dyes adsorption by prepared the HEC/PANI-PPy composite.  

Answer: The scheme showing proposed mechanism has been included in the manuscript along with the discussion.

Comment 6: How the pH is controlled at certain point to investigate the effect of pH, since the pH is important parameter to know the adsorption mechanism in this research. Is that the initial pH or equilibrium pH on the adsorption processes? The Authors should explain more even in the others effect on the adsorption process.  

Answer: The pH of the solution was adjusted with the known concentration of HCl and NaOH. The effect of pH was initial pH but after optimization of the pH, the rest of the study was performed at equilibrium pH. The other parameters which affect the adsorption were also studied such as the effect of contact time, dose of composite etc. which are given in the manuscript.

Comment 7: What is the electrical charge of surface sites of the HEC/PANI-PPy at different pH values?

Answer: The electrical charge of the surface sites of an adsorbent can generally be found from pH of the point of zero charge (pHpzc) but in the present study we were unable to perform that study.

Comment 8: The authors state the increasing the temperature will decrease the adsorption capacity, indicate endotherm process, but did not support the ΔH and others thermodynamic data. What is the possibility of chemical reaction and or ion exchange involve besides adsorption process?   

Answer: Many research studies indicate that if the adsorption capacity decreases with temperature, then the adsorption will be physical adsorption. The decrease in adsorption capacity with the rise in temperature is usually due to a decrease in the availability of active sites and the force of attraction between adsorbate and adsorbent

Comment 9: How about the reusability of this composite in adsorption-desorption process of dyes? What was the desorption efficiency (%)? This should be at least mentioned in the text. Would the same desorption method be used for dye degradation and reuse, or would a specific desorption method need to be developed for each contaminant?  

Answer: Of course, the regeneration and recycling of the composite is an important study but we were unable to perform this study. We hope that a desorption study will be carried out in of next research study.

Reviewer 2 Report

In this work, the authors reported synthesis and characterization of biocomposite wherein copolymer of polyaniline (PANI) and Polypyrrole (PPY) was grafted onto hydroxyethyl cellulose (HEC). Further, adsorption properties of as-prepared composite were evaluated using textile dyes Rhodamine B (RhB) and methyl Orange (MO)- as model adsorbate. The author employed various characterization and measurement methods to evaluate the absorption efficiency level of the prepared biocomposite. And a considerable results have been obtained. Overall, the work is interesting. But before its further consideration, some issues need to be verified.

1. In the abstract, some data results need to be given. Should not only underline what you do.

2. Why the authors employed HEC/PANI-PPy, the authors should discuss the advantages n in the introduction.

3. For Figure 6, Figure 8 and Figure 21, the images are not clear, the authors should improve the resolution.

4. A repeat measurement experiment for the absorption effect shoule be carried.

5. Some novel characterization methods and mechanism explanations should be enriched by referring to and citing following related literatures: Polymer Testing 2017, 59, 371-376; Environmental Science and Ecotechnology 2020, 3, 100035; RSC advances 2018,  8 (71), 40813-40822.

6. There are a large number of grammar and format issues. The authors are suggested to polish the manuscript carefully. 

Author Response

General comment: In this work, the authors reported synthesis and characterization of biocomposite wherein copolymer of polyaniline (PANI) and Polypyrrole (PPY) was grafted onto hydroxyethyl cellulose (HEC). Further, adsorption properties of as-prepared composite were evaluated using textile dyes Rhodamine B (RhB) and methyl Orange (MO) as model adsorbate. The author employed various characterization and measurement methods to evaluate the absorption efficiency level of the prepared biocomposite. And a consideration, some issues need to be verified.

Answer: Thank you very much for your favorable recommendation of our work. We have tried to further improve the quality of our manuscript by revising it according to your very logical comments.     

Comment 1: In the abstract, some data results need to be given. Should not only underline what you do.

Answer: Data results have been incorporated in Abstract

Comment 2: Why the authors employed HEC/PANI-PPy, the authors should discuss the advantages n in the introduction.

Answer: The needful has been done to highlight the advantages.

Comment 3: For Figure 6, Figure 8, and Figure 21, the images are not clear, the authors should improve the resolution.  

Answer: Figures have been improved

Comment 4: A repeat measurement for the absorption effect should be carried. 

Answer: Average of triplicated results were plotted.

Comment 5: Some novel characterization method and mechanism explanations should be enriched by referring to and citing following related literatures: Polymer Testing 2017, 59, 371-376; Environmental Science and Ecotechnology 2020, 3, 100035; RSC advances 2018, 8(71), 40813-40822.

Answer Suggested references have been incorporated

Comment 6: There are a large number of grammar and format issues. The authors are suggested to polish the manuscript carefully.  

Answer: We have tried our level best to improve the manuscript and you will find this revised version better.

Reviewer 3 Report

The following interesting paper entitled “Synthesis and Characterization of Hydroxyethyl Cellulose Grafted with Copolymer of Polyaniline and Polypyrrole Biocomposite for Adsorption of Dyes” have studied the synthesis, characterization, and application of hydroxyethyl cellulose (HEC) grafted with a copolymer of polyaniline (PANI) and polypyrrole (PPy) for water treatment purpose. This study has novelty and the manuscript is well structured and reads well overall, although it will need a spelling check. I suggest this article be published after a major revision.

Comments:

1- First of all, I highly recommend the authors provide a simple but informative “Graphical Abstract” for this study to intuitively present the big picture at a glance.

2- The abstract should be included the important findings of the report. Please revise it accordingly.

3-The novelty statement part of the introduction is poorly written. Please rewrite it and fully explain why the authors decided to synthesize hydroxyethyl cellulose (HEC) grafted with a copolymer of polyaniline (PANI) and polypyrrole (PPy)?

 4- Some of the references in the introduction part are too old (e.g., 1993, 1996, or 1998) and it is not acceptable at all. A myriad of research bodies has been published in recent years and you can find similar concepts and cite them in your paper rather than more than 3 decades old references. Moreover, in the introduction part, please read and add valuable information from the following key paper as well:  

(Conductive polymers in water treatment: A review = https://doi.org/10.1016/j.molliq.2020.113447)

(Interface Science and Technology Volume, Adsorbent = https://doi.org/10.1016/B978-0-12-818805-7.00009-6 )

(Nanoadsorbents based on conducting polymer nanocomposites with main focus on polyaniline and its derivatives for removal of heavy metal ions/dyes: A review = https://doi.org/10.1016/j.envres.2017.12.025)

(Conductive Polymers and Their Nanocomposites as Adsorbents in Environmental Applications = https://doi.org/10.3390/polym13213810)

5- Please repeat all experiments with at least three replications and provide all graphs with an error bar to increase the validity of the presented data ()

6- Reusability of adsorbents is of significant importance for their real-world application. Please provide reusability data for your material with at least 3-5 cycles.

7- The adsorption isotherm graphs are too low quality (it is not legible). Please swap them with better graphs.

8- To understand how temperature can impact the efficiency of the adsorbent, please provide adsorption thermodynamic data.

Author Response

General comment: The following interesting paper entitled “Synthesis and Characterization of Hydroxyethyl Cellulose Grafted with Copolymer and Polypyrrole Biocomposite for Adsorption of Dyes” have studied the synthesis, characterization, and application of hydroxyethyl cellulose (HEC) grafted with a copolymer of polyaniline (PANI) for water treatment purpose. This study has novelty, and the manuscript is well structured and reads well overall, although it will need a spelling check. I suggest this article be published after a major revision.               

Answer: Thank you very much for your favorable recommendation to our work. We have tried to further improve the quality of our manuscript by revising it according to your insightful comments.

Comment 1: First of all, I highly recommend the authors provide a simple but informative “Graphical Abstract” for this study to intuitively present the big picture at a glance.

Answer: This recommendation is really a valuable. We have added graphical abstract to present overall picture at glance.

Comment 2: The abstract should be included the important findings of the report. Please revise it accordingly.

Answer: Thanks for suggestion and abstract has been updated.

Comment 3: The novelty statement part of the introduction is poorly written. Please rewrite it and fully explain why the authors decided to synthesize hydroxyethyl cellulose (HEC) grafted with a copolymer of polyaniline (PANI) and polypyrrole (PPy)?

Answer: Thanks for insightful reading of the manuscript. We have tried our level best to modify the introduction part. 

Comment 4: Some of the references in the introduction part are too old (e.g., 1993, 1996, or 1998) and it is not acceptable at all. A myriad of research bodies has been published in recent tears and you can find similar concepts and cite them in your paper rather than more than 3 decades old references. Moreover, in the introduction part, please read and add valuable information from the following key paper as well:(Conductor polymers in water treatment: A review = https://doi.org/10.1016/j.molliq.2020.113447)

(Interface Science and Technology Volume, Adsorbent = https://doi.org/10.1016/B978-0-12-818805-7.00009-6)

(Nanoadsorbents based on conducting polymer nanocomposites with main focus on polyaniline and its derivatives for removal of heavy metal ions/dyes: A review = https://doi.org/10.1016/j.envres.2017.12.025)

(Conductive Polymers and Their Nanocomposites as Adsorbents in Environmental Applications = https://doi.org/10.3390/polym13213810)

Answer: In connection with recent developments, your suggestion is valuable. We have rescinded older references and added the latest references as instructed by you.

Comment 5: Please repeat all experiments with at least three replications and provide all graphs with an error bar to increase the validity of the presented data

Answer: Average of triplicated results were plotted.

Comment 6: Reusability of adsorbents is of significant importance for their real-world application. Please provide reusability data for your material with at least 3-5 cycles.

Answer: Of course, the regeneration and recycling of the composite is an important study, but we were unable to perform this study. We hope that desorption study will be carried out in of next research study.

Comment 7: The adsorption isotherm graphs are too low quality (it is not legible). Please swap them with better graphs.

Answer: Thanks for this valuable comments. Graphs are now improved and are legible.

Comment 8: To understand how temperature can impact the efficiency of the adsorbent, please provide adsorption thermodynamic data.

Answer: Many research studies indicate that if the adsorption capacity decreases with temperature, then the adsorption will be physical adsorption. The decrease in adsorption capacity with the rise in temperature is usually due to a decrease in the availability of active sites and force of attraction between adsorbate and adsorbent.

Round 2

Reviewer 1 Report

After reading carefully, I suggest this paper for publication in this journal.

Reviewer 3 Report

The manuscript is well-amended and ready for publication. I have no further comments.